# The Occurrence of Cattle Tick Fever in a Region of the Atlantic Forest on the Border with the Caatinga in Brazil

**DOI:** 10.3390/ani13233636

**Published:** 2023-11-24

**Authors:** Juan Dario Puentes, Vitor Santiago de Carvalho, Lais Gouveia Caymmi, Múcio Fernando Ferraro de Mendonça, Franklin Riet-Correa

**Affiliations:** 1Postgraduate Program in Animal Science in the Tropics, Federal University of Bahia, Salvador 40170-110, Brazil; lais_caymmi@hotmail.com (L.G.C.); franklinrietcorrea@gmail.com (F.R.-C.); 2Livestock Development Center, Federal University of Bahia, Santo Amaro 44200-000, Brazil; vitornet@gmail.com

**Keywords:** anaplasmosis, babesiosis, bovine, Brazil, cattle movement, *Riphicephalus (Boophilus) microplus*

## Abstract

**Simple Summary:**

In this report, some epidemiological aspects of cattle tick fever in an area of the Atlantic Forest biome bordering the Caatinga biome are described. A survey regarding cattle tick fever was carried out in farms visited by the Livestock Development Center of the Federal University of Bahia. Tick fever was the most frequent disease diagnosed in the region, even though it is an endemic area. The disease mainly occurred in two groups of animals: adult animals moved from the Caatinga, a non-endemic biome with a semi-arid climate, and calves who were born in the Atlantic Forest, an endemic biome, but subjected to high tick loads before they were nine months old. The disease occurred mainly in dairy *Bos taurus* or *B. taurus* crossbred cattle, and the most frequent etiologic agent was *Anaplasma marginale*. The control and prophylaxis of the disease should focus on cattle moved from the Caatinga to the Atlantic Forest biome, as well as on dairy cattle residing in the area.

**Abstract:**

The Atlantic Forest biome is considered an area in which tick fever is endemic, while the Caatinga biome is considered an area in which tick fever is non-endemic. A survey on cattle tick fever was carried out in 448 cattle farms located in an area of the Atlantic Forest biome which borders the Caatinga biome. A retrospective study of five years, conducted from January 2018 to October 2022, allowed for the identification of the occurrence of tick fever in 44 out of the 448 herds visited. In 70.5% (31/44) of the herds involved, the disease was caused by *Anaplasma marginale*; in 6.8% (3/44), the disease was caused by *Babesia* spp.; and in 22.7% (10/44), the disease was caused by a coinfection of *Babesia* spp. and *A. marginale*. The disease mainly occurred from August to November (23/44). *Bos taurus* or *B. taurus* crossbreed animals were most affected (29/44) in an area in which 94% of the cattle were *Bos indicus* and 6% were *B. taurus* and *B. taurus* crossbreeds. In 24 herds (with adults affected in 17 and calves in 7), the implicated animals had recently been moved to the Atlantic Forest. In the other 20 herds (calves with heavy tick infestations were affected in 17 and adults in 3), tick fever occurred in non-moved cattle. Even though it is an endemic zone, tick fever is common and mainly affects *B. taurus* cattle, including adults moved from areas with enzootic instability and calves under nine months old with high tick loads.

## 1. Introduction

Cattle tick fever is a complex of diseases caused by *Anaplasma marginale* (order: Rickettsiales), an alphaproteobacteria [1], and *Babesia bigemina* and *Babesia bovis*, apicomplexa protozoa [2]. These agents are mainly transmitted biologically by the tick *Rhipicephalus (Boophilus) microplus* [3,4]. They may be also transmitted via the transplacental route [5]. Additionally, *A. marginale* can be transmitted mechanically by bloodsucking flies of the genera *Tabanus* and *Stomoxys* or by needles and other blood-contaminated fomites [6]. Bovines with tick fever have anorexia, somnolence, weakness, dry feces, tachypnea, fever, hemolytic anemia, and jaundice, which are followed by death if not treated. In the case of babesiosis, hemoglobinuria is also observed, and in *B. bovis* infections, nervous signs may occur [7]. If calves become infected for the first time before nine months of age, they may acquire immunity without becoming sick. The infected calves will become asymptomatic chronic carriers and, if they are later infected as adults, they will not develop acute disease [7,8]. However, if calves under nine months of age become infested with large numbers of ticks infected with *A. marginale* or *Babesia* spp., they can develop tick fever [9].

A region with an instability situation with respect to babesiosis is one in which the number of ticks infesting hosts per day is too low for calves under nine months of age to become infected with *Babesia* spp. [10]. The Caatinga biome in Brazil was previously categorized as an area of enzootic instability for cattle tick fever because the low humidity impairs tick development [11]. A disease is considered endemic if it is more frequent in older individuals than in younger ones and if the initial infection reduces the probability that subsequent infections result in acute presentations [12]. This stable tick fever situation has been documented in the Atlantic Forest biome in Brazil, where the environmental conditions are ideal for tick development [11]. The occurrence of cattle tick fever may be higher in the border region between the Caatinga and the Atlantic Forest biomes due to animal movement, limited surveillance, and poor management conditions.

Tick fever can cause high economic losses because of mortality, weight losses, subclinical productive losses, and the cost of treatments. In the State of Rio Grande do Sul, an important region for the production of livestock which is commonly perceived to have enzootic instability for cattle tick fever [11], the disease causes losses due to mortality, which is estimated at 1.6 million dollars annually [13].

In the different environmental conditions of Brazil, retrospectives studies of cattle tick fever have been performed with the aim of estimating the relevance and describing some epidemiological aspects of the disease [13,14,15]. However, there is a lack of information regarding the disease’s ecology in the border region between the Caatinga and the Atlantic Forest biomes. The aim of this study was to describe some epidemiological aspects of tick fever in cattle farms from an area of the Atlantic Forest biome bordering the Caatinga biome in the State of Bahia, Brazil.

## 2. Materials and Methods

We carried out a survey of cattle tick fever in 448 farms from an area of the Atlantic Forest which borders the Caatinga biome. The size of the herds ranged from 5 to 1020 animals. The farms were visited by the Livestock Development Center of the Federal University of Bahia (CDP-UFBA) from January 2018 to October 2022. The CDP-UFBA is located in the Atlantic Forest biome, and farms from the municipalities in the area bordering the Caatinga biome were visited when required because of the occurrence of diseases in cattle. In this region of the Brazilian State of Bahia, 79.7% of farmers (10,886 of 13,661) are small producers who practice subsistence farming [16]. The climate of the Atlantic Forest in this region is predominantly humid tropical. The mean temperature is 26 °C, and the mean rainfall is 2500 mm per year, with the highest volume of precipitation occurring from March to August, during the rainy season [17].

This study was carried out on farms located in the Atlantic Forest biome bordering the Caatinga biome that presented a cattle disease diagnosed as tick fever (Figure 1). In each selected herd, information regarding clinical signs, morbidity and mortality, the etiologic agent, the date of occurrence, the age and breed of the affected animals, and the municipality in which tick fever occurred was collected. Because the observation of *A. marginale* and/or *Babesia* spp. in the peripheral blood smear of animals in an endemic area does not necessarily indicate the occurrence of tick fever, an animal was diagnosed with acute cattle tick fever when it met all of the following criteria: infestation with ticks; clinical signs of lethargy, weight loss, fever, pale mucous membranes, jaundice, and dry feces; a packed cell volume (PCV) lower than 22%; a red blood cell count (RBC) lower than 5.1 × 10^6^ cells per microliter [18]; the presence of *A. marginale* and/or *Babesia* spp. in the peripheral blood smear; and the improvement of the clinical signs after treatment (blood transfusion; the subcutaneous administration of a unique dose of 3 mg/kg of imidocarb dipropionate in the case of babesiosis and/or 11 mg/kg of intravenous oxytetracycline once a day for five days in the case of anaplasmosis). An animal was considered to have a high tick infestation if it was infested with 22 or more ticks (measuring 4.5–8 mm in length). The number of parasites was determined by counting the ticks on one side and multiplying the result by two [19].

The frequencies were analyzed using descriptive statistics. Since calves have an innate immune response to tick fever that lasts until nine months of age [7,8], the herds diagnosed with tick fever were split into two categories based on the age of the affected animals: young (under nine months old) and adult (more than nine months old). The sums of morbidities and mortalities were averaged, and the standard deviation (SD) was calculated for each one.

## 3. Results

Between January 2018 and October 2022, tick fever was diagnosed in 44 out of the 448 herds visited by the CDP-UFBA. In 24 herds involved, the mean morbidity was 27.8 ± 31.2% (mean ± SD), and in 16 herds, the mean mortality was 10.8 ± 13.6%. No morbidity or mortality data were available in other herds (Appendix A). In 70.5% (31/44) of the herds, the disease was caused by *A. marginale*; in 6.8% (3/44), the disease was caused by *Babesia* spp., and in 22.7% (10/44), the disease was caused by a coinfection of *Babesia* spp. and *A. marginale*. In 52.3% (23/44) of the herds, tick fever occurred from August to November every year (Figure 2). In 54.5% (24/44) of the herds, the disease occurred in bovines under nine months of age and in 45.5% (20/44) of adult cattle. In 65.9% (29/44) of the herds, the affected animals were *Bos taurus* or a *B. taurus* crossbreed, and 34.1% (15/44) were *Bos indicus*. The herds involved belonged to the municipalities presented in Figure 1.

In 24 herds affected by tick fever, it was reported that the affected animals had been recently moved to the Atlantic Forest biome; however, information about the animals’ origins was obtained for only 10 herds; in six of these herds, the animals had recently been moved from the Caatinga biome to the Atlantic Forest biome; in three, the cattle had been recently moved between different areas within the Atlantic Forest biome; and in one, the cattle were carried from a pen to a pasture where they became infested with ticks. In 14 out of the 24 herds in which cattle had been recently moved, the disease occurred in adult animals, and it only occurred in animals under nine months of age in 3 herds. In contrast, in 17 out of the 20 herds in which tick fever was not associated with cattle movement, the disease occurred in animals under nine months of age that were infested with more than 22 teleogines and only occurred in adults in 3 herds.

## 4. Discussion

The results of the present study suggest that tick fever is prevalent on cattle farms in the Atlantic Forest biome on the border with the Caatinga biome in the State of Bahia, Brazil. We based our diagnosis on the observation of clinical signs and blood parameters characteristic of tick fever. However, it could not be ruled out that some of the animals within the involved herds may have been co-infected with other pathogens responsible for similar clinical signs, considering that parasite and bacteria counts were not performed as criteria for the diagnosis of anaplasmosis and babesiosis.

In non-endemic areas of Brazil, tick fever has been frequently diagnosed by veterinary diagnostic laboratories. In the Caatinga biome of the State of Paraíba, the disease represented 1.3% (14/1113) of bovine diagnoses [20] and in the Pampa biome of the States of Santa Catarina and Rio Grande do Sul, it represented 6.9% (112/1623) and 5.6% (328/5887) of bovine diagnoses, respectively [21,22]. In contrast, in endemic areas, the frequency of the diagnosis of tick fever by veterinary diagnostic laboratories is lower. In the Cerrado biome of the States of Mato Grosso do Sul and Mato Grosso, the frequencies of the diagnosis of cattle tick fever were 0.35% (19/5298) and 1.6% (9/554), respectively [23,24]. Previous studies in cattle from the Atlantic Forest biome in the State of Bahia reported high rates of infection with *A. marginale* and *Babesia* spp. transmitted by ticks [25], so this region has been identified as an endemic area [11,26]. Despite this, our study found a high frequency of tick fever. The recent movement of cattle, mainly of adult cattle, was a common feature present in various herds in which the CDP-UFBA diagnosed tick fever, especially, in those herds in which adult animals were moved from the Caatinga biome to the Atlantic Forest biome. In the Caatinga biome, where the climate is semi-arid, the low-humidity conditions limit the development of *R. microplus*, resulting in the enzootic instability of tick fever. In the Atlantic Forest, the climatic conditions are optimal for tick development and, consequently, *A. marginale* and *Babesia* spp. circulate among animals frequently [11], with the persistently infected animals being the source of infection [27]. Cattle moved from the Caatinga biome to the Atlantic Forest biome may be susceptible to tick fever because they come from an area with few or no ticks and, consequently, they may have low levels of antibodies against *A. marginale* and/or *Babesia* spp.

Tick-infested calves rarely display clinical signs of anaplasmosis [8] or babesiosis [7]. In the case of babesiosis caused by *B. bovis,* it is assumed that calves do not exhibit clinical signs of the disease because of their increased nitric oxide production by spleen cells [28]. Due to this innate protection, calves from endemic areas in which ticks are present for the whole year can develop immunity without becoming ill when they acquire the infection [29]. However, tick fever in calves from endemic zones has been associated with high levels of tick infestation [23,30]. In this study, the majority of tick fever diagnoses unrelated to cattle movement were performed on bovines under nine months of age with high levels of tick infestation, particularly towards the end of the rainy season between August and November, when tick loads are the highest.

The breed of the affected cattle and the type of production system were related to the occurrence of tick fever in this study. Even though in the studied region, it was determined that 94% of the cattle were *B. indicus* and only 6% were *B. taurus* and *B. taurus* crossbreeds [16], in most of the herds diagnosed with tick fever (65.9%), the animals affected were *B. taurus* or *B. taurus* crossbreeds. A similar situation was reported in other regions of Brazil [13,14,15]. Bovines of European breeds are more susceptible to tick infestation and tend to be more susceptible to babesiosis [7], while *B. indicus* cattle resist *R. microplus* infestations better [31]. In the studied region, the introduction of *B. taurus* breeds to improve the profit of bovine farms could be an important predisposing factor in the occurrence of tick fever.

In the present study, cattle tick fever occurred throughout the year; however, most infections occurred at the end of the rainy season and the beginning of the dry season, from August to November. In the studied region, the highest tick infestations in cattle were detected from September to January, depending on the amount of rainfall [11]. A similar situation was reported in the Caatinga biome of the State of Paraíba, where tick fever is more frequent during the rainy season and the beginning of the dry period when tick loads are higher [14]. In the Atlantic Forest, most cases of tick fever occurred towards the end of the rainy season when the humidity and temperature conditions are adequate for the development of *R. microplus* [11].

These findings suggest that the control and prophylaxis of the disease should be focused on cattle being moved from the Caatinga biome to the Atlantic Forest biome, as well as in dairy cattle residing in the region. To prevent tick fever in cattle moved from areas with enzootic instability, such as the Caatinga biome, it would be necessary to immunize them with vaccines. However, currently, there are no available vaccines against *A. marginale* and/or *Babesia* spp. in Brazil. As an alternative, the administration of preventive doses of long-acting tetracycline for *A. marginale* [29] and imidocarb for *Babesia* spp. [32] during the acute infection has been suggested. However, this should be considered with caution as it is known that the administration of imidocarb can result in prolonged withdrawal times in the tissues of treated animals and, although adverse reactions in humans have not been reported, its use in cattle intended for human consumption could pose a risk to human health [33]. It also should be considered that the preventive treatment could favor antibiotic resistance as genes associated with oxytetracycline resistance have been found in *A. marginale* isolates [34]. To prevent the occurrence of tick fever in cattle under nine months of age residing in endemic regions such as the Atlantic Forest biome, it would be important to diminish the loads of ticks infesting animals through the use of acaricides, especially during the period of higher infestations. However, since tick resistance to amitrazin, organophosphates, pyrethroids, and organophosphate-pyrethroid combinations has been reported in nearby regions [35], evaluating the efficacy of acaricides before executing any control plan is recommended. In the future, it will be necessary to conduct studies on the dynamics of the tick life cycle, including the number of generations in the region, to develop suitable control measures for ticks and tick fever. Dairy farmers who decide to introduce *B. taurus* cattle or their crossbreeds to improve their farm’s profitability should consider the high susceptibility of these types of animals to *R. microplus* infestation and, therefore, the increased risk of tick fever.

In this study, *A. marginale* was identified as the most frequent agent causing tick fever in herds in the Atlantic Forest biome on the border with the Caatinga biome. In other studies of endemic and non-endemic regions of Brazil, *A. marginale* also was the most frequent agent diagnosed, probably because this bacterium is transmitted by a great variety of vectors and fomites [14,15]. However, in Southern Brazil, *B. bovis* was identified as the most frequent agent causing tick fever [22]. In the area of influence of this study, *R. microplus* appears to be the primary vector of *A. marginale.* However, it is not ruled out that blood-sucking Diptera of the genera *Tabanus* and *Stomoxys* or blood-contaminated fomites like needles, dehorning saws, nose rings, tattoo machines, earrings, and castration instruments may play a role in the transmission of *A. marginale* [6]. For this reason, it is necessary to investigate the relevance of these means of the transmission of *A. marginale* in the transition area between the Atlantic Forest and the Caatinga biome in Brazil.

## 5. Conclusions

Tick fever is an important disease that affects cattle in an area of the Atlantic Forest biome on the border with the Caatinga biome. The disease primarily affects *B. taurus* or *B. taurus*-crossbred cattle that have been recently moved to the Atlantic Forest biome, particularly if they come from the Caatinga biome, or in bovines under nine months of age, mainly from September to November, which is a period of higher tick loads. This information should be useful for better comprehension and management of tick fever in regions with comparable conditions.

## Figures and Tables

**Figure 1 animals-13-03636-f001:**
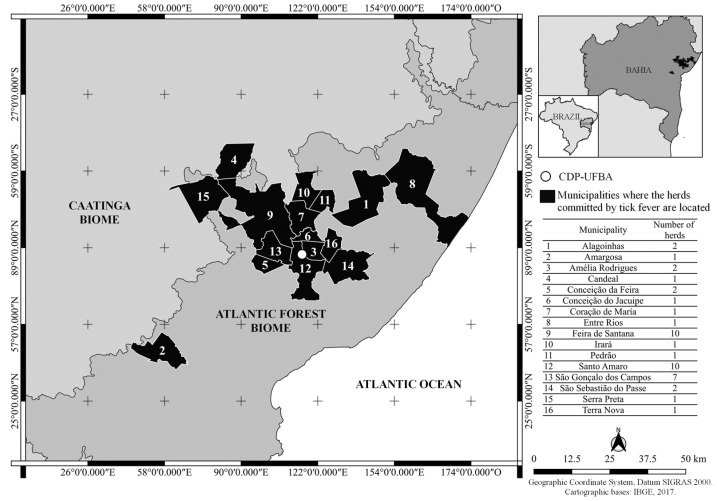
Municipalities of the herds in which tick fever was diagnosed by the CDP-UFBA between January 2018 and October 2022. In the map, the white circle indicates the CDP-UFBA’s localization, and each municipality is identified with a corresponding number listed in the table adjacent to it. The table displays the names of the municipalities and the corresponding number of herds involved in each municipality.

**Figure 2 animals-13-03636-f002:**
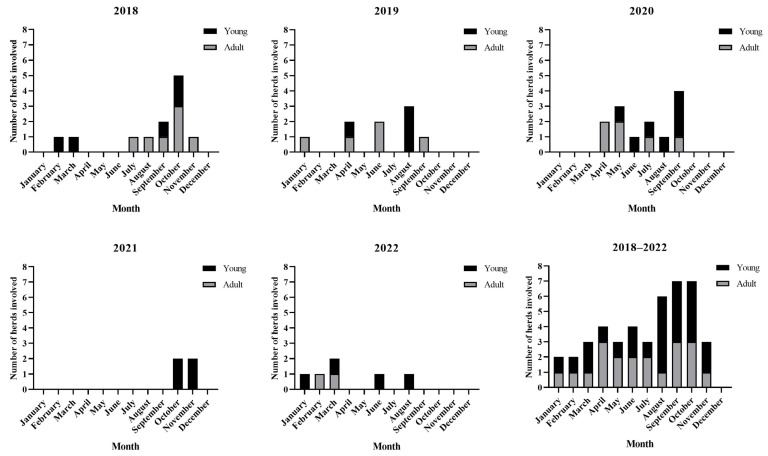
Monthly distribution of tick fever occurrence in 44 of the 448 farms visited by the CDP-UFBA between 2018 and 2022. The first five bar graphs in the panel represent each individual year studied. The final bar graph categorizes the entire period analyzed. The number of herds involved by month are represented by vertical bars. The black side of the bar represents the number of herds in which the affected animals were bovines under nine months of age and the gray side represents the number of herds in which the affected animals were bovines older than nine months.

## Data Availability

Data will be made available upon request.

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
