# Peer review of "The Occurrence of Cattle Tick Fever in a Region of the Atlantic Forest on the Border with the Caatinga in Brazil"

_animals, 2023, doi:10.3390/ani13233636_

Round 1

Reviewer 1 Report (Previous Reviewer 2)

Comments and Suggestions for Authors

Dear authors,

Here are my suggestions and comments:

- Line 24: Replace "5" with "five".

- Line 26: Replace "spp" with "spp.".

- Line 29: Replace "7" with "seven".

- In the abstract, replace "Bos taurus" repetitions with "B. taurus".

- Line 31: Replace "3" with "three".

- Line 67: Replace "15] In" with "15]. In".

- Line 67: When mentioning the damage in Rio Grande do Sul, mention that it is also an area of enzootic instability, as this justifies the insertion of the data.

- The final paragraph of the introduction looks like a set of isolated sentences that do not dialogue with each other. This text needs more fluidity.

- Line 99: Replace "and / or" with "and/or".

- In the results, replace "Bos taurus" repetitions with "B. taurus".

- Line 131: Replace "6" with "six".

- Line 132: Replace "3" with "three".

- Line 133: Replace "1" with "one".

- Line 136: Replace "3" with "three".

- Line 138: Replace "3" with "three".

- The conclusions can be better elaborated for a wider application on the enzootic instability of the diseases than regionalized.

The study is much better grounded and written than the first time. I congratulate the authors and wish them success.

Kind Regards.

Author Response

Reviewer 2 Report (Previous Reviewer 3)

Comments and Suggestions for Authors

The manuscript has been improved overall, although I still found some aspects that require additional discussion. One of these points is the lack of bacteremia and parasitemia counts and the other important one is that the use of antimicrobials should not be suggested as a preventive measure.

Introduction

Line 11: please use “we describe” instead of “are described” and  replace “epidemiologic” for  “epidemiological”

Line 13: please use “was carried” instead of “were carried”

Line 15: use “groups of animals” instead of “situations”

Line 18: “mainly” instead of “principally”

Material and methods

Line 85-87: I think this sentence requires rewriting, mainly the way it starts, maybe: “For the study we selected those herds that…”

Line 93: replace “parameters” by “criteria”

Line 96: what about parasitemia count? you can get that information from the blood smears, is an important data to correctly define an outbreak. I think that this is an important missing data.

Line 97: please use “history of” instedad of “antecedents of”

Results

Line 118: please replace “belongs” with “belong”

Discussion

Lines 144-145: I do not agree with this statement “the observation of clinical signs and blood parameters characteristic of tick fever allow an accurate diagnosis of this disease”. I would rather highlight that this aspect is a great weakness of the study.

Line 146: use “responsable for similar clinical signs” instead of “that exhibit similar behavior”

Line 158: use “translocated” or “moved from” instead of “carried”

Lines 170 to 174: It is not clear in this pharagraph if calves do or do not show clinical signs. You begin mentioning that they do show clinical signs but then you talk about the innate inmune that protect calves from clinical disease.

Lines 177 and 178: please replace “are” by “were”

Line 187: please replace “occurs” by “were detected”

Line 188: please use “was” instead of “is”

Line 190: please use “higher” instead of “heavier”

Line 191: “occured” instead of “occur”

Lines 198-199: the authors still suggest the use of antimicrobials as preventive treatment. This is not a novel way to control the cattle tick fever and is a threat in terms of One-Health. I strongly suggest to discourage the use of antimicrobials as preventive treatment.

Line 209: erase the words “the of”

Line 215: use “great variety” instead of “high variety”

Line 222: use “ways” instead of “modes”

References

An additional reference regarding babesiosis and anaplasmosis in the state of Bahia is the following, I suggest to discuss it.

_ Amorim LS, Wenceslau AA, Carvalho FS, Carneiro PL, Albuquerque GR. 2014. Bovine babesiosis and anaplasmosis complex: diagnosis and evaluation of the risk factors from Bahia, Brazil. Rev Bras Parasitol Vet. 23(3):328-36. doi: 10.1590/s1984-29612014064.

Comments on the Quality of English Language

Editing of English language required

Author Response

Reviewer 3 Report (New Reviewer)

Comments and Suggestions for Authors

The submitted Communication manuscript describes a retrospective study of cattle tick fever aimed to describe some epidemiological aspects of the disease in cattle farms of the Atlantic Forest biome in the State of Bahia, Brazil. 

Studies aimed to estimate the infection rates of tick fever in cattle in a particular country or region are usually performed as part of the epidemiological surveillance conducted by the veterinary services office of the regulatory authorities in charge. As such, the manuscript presented is not highly original. The novelty of this study, perhaps, relies on the approach used to gather the data. By combining the availability of records obtained from the area of influence for the Livestock Development Center of the local university, and the data analysis of the survey about cattle tick fever that was carried out by visiting, during a period of 5 years (2018-2022) 448 cattle farms located in an area of the Atlantic Forest biome in Bahía, Brazil. Such visits allowed authors to identify and stratify, according to the age of affected animals, cattle tick fever occurring in 44 herds out of the 448 farms visited.

Thus, the results could be of clinical veterinary importance as it is clearly shown that a substantial proportion of farms presented clinically affected cattle infected with the economically important tick-borne disease caused by Anaplasma marginale, and Babesia spp. The results obtained are thus of primary interest to the local authorities and cattle farmers, but could also be of regional importance for the cattle industry in Northern Brazil and provide with additional data that will contribute to the knowledge and understanding of the global distribution of tick-borne pathogens. Overall, the manuscript is well written, the laboratory microscopic detection methods utilized are adequate, and the technical details are provided to replicate the work. However, although providing sound and useful information, the paper should be improved before being published.

Simple Summary

Line 11. Rephrase sentence to “In this report,  some epidemiologic aspects of the cattle tick fever in  an area of the Atlantic Forest biome, bordering the Caatinga biome are described.”

Line 12. Correct sentence to “A survey about cattle tick fever was carried out in farms visited by…”

 Abstract

Line 22.  Correct sentence to “A survey about cattle tick fever was carried out in 448 cattle farms…”

Lines  24-25.  Correct phrase to “A retrospective study of 5 years, conducted from January 2018 to October 2022, allowed to identify tick fever occurrence in 44 herds, out of 448 visited.”

Line 25. Use “involved” or “visited”, instead of “committed”. Check and replace terms throughout the manuscript (see lines 109, 110,111, 114, 116, 118, 120, 127, 129, 131, 132, 133).

 Line 33. It is mentioned that “…cattle under nine months old with high tick loads.”. Authors should describe what a “high tick load” means, as there is no indication of quantitative referral to tick counts in affected animals in the whole manuscript.

Introduction

Line 44. In describing the clinical signs in cattle associated with tick fever, authors can use “anorexia and dumless", instead of “loss of appetite and obtundation”.

2. Materials and Methods

Line 74. It is mentioned that “We carried out a survey about cattle tick fever in 448 farms from an área…”  What is the average herd size?… Authors should indicate the average or ranges of the herd size.

Line 85. Rephrase to “herds located in the Atlantic Forest bordering the Caatinga biome were selected for the study.

Lines 104-105. It is mentioned that “the sum of morbidities and mortalities were averaged…”. Please indicate the total number of diagnosed cases, and /or describe the total number of sick and dead animals out of the total number of animals in the 44 herds studied. Use “statistical” instead of “statistic”.

3. Results

Lines 109-110. It is mentioned that “In 24 herds committed, mean morbidity was 27.8%±6.4% (mean±SEM) and, in 16 herds committed, mean mortality was 10.8%±3.4%.” It was previously indicated that 44 farms/herds were included in the study. What about the other 4 farms?

Lines 130-131. Please rephrase “...the origin information was only available in 10: in 6 committed herds, the animals had been recently moved from the Caatinga biome”. As written, it is confusing.

 4. Discussion

Line 158. Use “those” instead of “that”

Line 203. Use “human consumption” instead of “human consume”.

Lines 208-209. Something is missing in the statement “Dairy farmers who decide to introduce Bos taurus cattle or their crosses to improve the of their...”

Comments on the Quality of English Language

In general, moderate editing of the English language is required

Author Response

Reviewer 4 Report (New Reviewer)

Comments and Suggestions for Authors

The paper by Puentes et al. describes an observational study of the prevalence of cattle tick fever in Brazil. It adds some value to the literature regarding the prevalence of cattle tick fever in the country. However, some of the methods are not clear. For example, what does heavy tick infestation mean? What tick counts are categorized as low infestation and what tick counts mean heavy infestation? Some more minor suggestions are mentioned below.

Simple summary

Line 13: was carried…

Abstract

Did the disease occur between August to November every year?

What is heavy tick infestation? (please provide counts)

Introduction

Line 52: ‘where’ instead of ‘were’

Line 56: if it is more…

Line 68: ‘ecology’ instead of ‘behavior’

Materials and Methods

Line 74: survey of cattle fever

Line 85 – 87: I think this sentence is redundant and can be deleted.

Line 109: I think the word ‘committed’ can be removed throughout this paragraph.

Did the authors make an estimate of tick counts in the two biomes through tick drags or by any other means? Or did they count the number of ticks on the animals? Presenting the tick numbers would be beneficial.

Discussion

Line 140: I think the starting sentence of discussion should instead say…. suggest that cattle fever is prevalent on cattle farms…. (instead of cattle fever has a significant impact on….) because you did not study the impact of cattle fever on animal health or yield in these farms.

Line 157: please remove ‘of the’ in between various herds…

Line 158: ‘those herds’ instead of ‘that herds’

Figures and Tables

Fig.2. should be presented as Fig.1. and referred to in the materials and methods section instead of in the results to give an idea of the general location of the regions to the readers.

Comments on the Quality of English Language

The quality of English language presented in this manuscript is good.

Round 2

Reviewer 4 Report (New Reviewer)

Comments and Suggestions for Authors

I have few suggestions mentioned below:

Discussion

The authors have only discussed the use of vaccines or drugs for tick fever management but haven't mentioned the use of acaricides for tick control in this region. Please elaborate on the chemical and cultural methods of tick management as well.

Figures and Tables

Fig. 2 should be shown by year. So it should have a panel of 5 bar graphs which would show a more detailed picture of the seasonality of tick fever occurrence in this region.

Round 3

Reviewer 4 Report (New Reviewer)

Comments and Suggestions for Authors

Line 61: ‘more’ instead of ‘increased’

Line 136: ‘affected by’ instead of ‘involved by’

Line 220: ‘during the period’ instead of ‘from the period’

This manuscript is a resubmission of an earlier submission. The following is a list of the peer review reports and author responses from that submission.

Round 1

Reviewer 1 Report

Comments and Suggestions for Authors

This study presents novel data on the epidemiology of bovine anaplasmosis and babesiosis in the Atlantic Forest and Caatinga biomes of the State of Bahia, Brazil. The work is well written, and I consider it can be published.

Minor suggestions:

Line 21. CHANGE: bred cattle and the most frequent etiologic agent causing the outbreaks was A. marginale. FOR: bred cattle and the most frequent etiologic agent causing the outbreaks was Anaplasma marginale.

Lines 27 to 30. CHANGE: The 70.5% (31/44) of the tick fever outbreaks were caused by A. marginale; 6.8% (3/44), by B. bigemina; 18.2% (8/44), by B. bigemina and A. marginale coinfection; and 4.5% (2/44), by B. bovis and A. marginale coinfection. The outbreaks occurred mainly from August to November (23/44). FOR: The 70.5% (31/44) of the tick fever outbreaks were caused by Anaplasma marginale; 6.8% (3/44), by Babesia bigemina; 18.2% (8/44), by B. bigemina and A. marginale coinfection; and 4.5% (2/44), by Babesia bovis and A. marginale coinfection. The outbreaks occurred mainly from August to November (23/44).

Line 219. CHANGE: Cattle Fever Tick Rhipicephalus Annulatus FOR: Cattle Fever Tick Rhipicephalus annulatus

Line 230. CHANGE: Babesia Bigemina e Babesia Bovis FOR: Babesia bigemina e Babesia bovis

Line 259. Idem for line 230.

Comments on the Quality of English Language

Minor editing of English language required.

Reviewer 2 Report

Comments and Suggestions for Authors

Dear authors,

here are some comments and suggestions:

- The manuscript deals with a very pertinent subject, which causes economic and health damage to production animals, and the study is interesting for readers of the journal. However, the way the study is approached diminishes its relevance. There is a lot of emphasis in the Simple Summary, in the Abstract and in the main text, about Bahia. The title points to the fact that it is an Atlantic Forest region, bordering the Caatinga, and this aspect is much more important than being reports from Bahia. Explore the biomes more (one endemic, the other non-endemic), explore the epidemiological history, more than the regional aspects of the study. The transition of the biomes is greater than Bahia.

- Why approach in the methodology that a review of cases was done? When I read this, the impression I have is that: I took some attendance sheets, took advantage of it and saw what I had. Instead of "reviewing cases", the study could have a greater impact if "we carried out a survey about cattle tick fever in 448 animals from a transition area of the Atlantic Forest and Caatinga". Do you understand that the way I write the methodology decreases or increases its relevance?

- In the introduction, better background on why outbreaks occur in the area.

- In the methods, describe the factors analyzed for the diagnosis of cattle fever tick. This is simplistically described in the results, but should be in the methods.

- The absence of a more accurate description of diagnostic techniques, in relation to the detection of pathogens, is a weakness of the study.

- The study is relevant. It has great figures and the efforts of the authors to bring better approaches to the data is remarkable, but the study needs to mature in writing. I really hope the comments help to bring out the best version possible.

Minor comments:

- Line 21: Replace "A. marginale" with "Anaplasma marginale".

- Line 28: Replace "A. marginale" with "Anaplasma marginale".

- Line 28: Replace "B. bigemina" with "Babesia bigemina".

- Line 29: Replace "B. bovis" with "Babesia bovis".

- Keywords repeat title terms, which should be avoided.

Reviewer 3 Report

Comments and Suggestions for Authors

Review_report

The study aims to analyze the situation of bovine anaplasmosis and babesiosis in the State of Bahia, Brazil. Eventhough this information is important to further prevention and control meassures, there are many inaccuracies. Is not clear whether an outbreak is a case (a bovine) or a herd with positive bovines. The diagnostic tools used are not accurate enought and other differential diseases were not analyzed. Hereinafter the discussion of results and conclusions are not well supported by the methodology used.

Abstract

Line 36: what do you mean with “unstable zones”? are you talking about areas with enzootic instability? If yes, please use this last term.

Introduction

Lines 41-42: References for Anaplasma marginale and Babesia species are crossed, Reference [1] is for Babesia bigemina and B. bovis. Reference [2] is the one for A. marginale

Line 43: Reference [3] is just for R. annulatus and for babesiosis transmission, please include another reference for R. microplus (could be: Gray, J.S., Estrada-Pena, A. and Zintl, A. (2019) Vectors of babesiosis. Annual Review of Entomology 64, 149–165) and any reference regarding to A. marginale transmission by these tick species.

Line 62: please replace “the objective of this study is” for “the aim of this study was”

Materials and methods

It is not clear to me if you are analyzing the herd as a unit, in the results section you describe the blood smear examination per animal, but then you conclude at an outbreak (herd?) level. Do you call outbreak to an acute case?

Line 76: please use “cases” instead of ”attendances” and “in which” instead of “where”

Line 78: which is the reference for these diagnostic criteria? (PCV and RBC cut-off points).

Parasitemia and bacteriemia were not calculated? In tick fever endemic regions persitently infected animals can present low babesia and anaplasma parasitemias that may obscure the diagnostic of other diseases. Parasitemia and bacteriemia levels in the context of an outbreak are well reported and should be taken in consideration for the outbreak definition. Some other diseases may present similar clinical signs and the solely detection of structures suggestive of Babesia or A. marginale in the blood smear could lead to misdiagnosis.

Line 80: use ”regarding” instead of “about”

Line 83-84: please replace "Because of the resistant condition of calves to tick fever" for: "Since calves have an innate immune response to tick fever that lasts until nine months of age”,

The authors only used  descriptive statistics

Results

Line 89: what do you mean with “concluded a diagnosis on 448 occasions.” Is that the diagnosis was confirm in the CDP-UFBA for 448 outbreaks/cases? Again, is very confusing because is not clear if you call outbreak to each case.

Lines 94-96: How did you discriminate between B. bovis and B. bigemina? Only blood smear examination was described in the materials and methods section and that methology is not enought to correctly discriminate between both Babesia species.

A table summarizing all the samples data (animal age, breed, origin, month, type of production,  parasite detection) is missing.

Line 113: please use “was available only for 10 outbreaks” instead of “was just mentioned in 10 outbreaks”

Line 114: Please use ”moved” instead of ” transported”

Line 114: please add “cases” after 3

Line 115: please add “case” after 1

Discussion

Lines 121-123: Other diseases with similar clinical signs (leptospirosis for example) were not tested, you can not rule out if coinfections or even a primary ethiologic agent different from Babesia or Anaplasma is responsable for the clinical signs. The blood smear examination (even more without parasitemia quantification) is not an accurate methodology to diagnose an outbreak by these diseases.

Line 135: you didn´t introduced the term “enzootic instability” previously, please exaplain this concept as is important to understand the idea explained further in lines 136-141.

Please use ”cattle movement” instead of “cattle transportation” all over the manuscript

Line 174-175: Is not clear whether the treatment propoused is preventive or for acute cases, if is the first option, I suggest to discuss this kind of indications in the context of One-Health and the increasing animicrobial resistance.

References

In the reference:  Puentes JD, Riet-Correa F. 2023. Epidemiological aspects of cattle tick fever in Brazil. Rev Bras Parasitol Vet. 32(1):e014422. doi: 10.1590/S1984-29612023007 the presence of cattle tick fever in the state of Bahia is mentioned but not discussed in this study.

An additional reference regarding babesiosis and anaplasmosis in the state of Bahia is the following, I suggest to discuss it.

_ Amorim LS, Wenceslau AA, Carvalho FS, Carneiro PL, Albuquerque GR. 2014. Bovine babesiosis and anaplasmosis complex: diagnosis and evaluation of the risk factors from Bahia, Brazil. Rev Bras Parasitol Vet. 23(3):328-36. doi: 10.1590/s1984-29612014064.

Comments on the Quality of English Language

I think that the english language requires improvement